# Remote ischemic preconditioning does not influence lectin pathway protein levels in head and neck cancer patients undergoing surgery

Kristine Frederiksen[1], Andreas Engel Krag[1,2,3], Julie Brogaard Larsen[1], Birgitte Jul Kiil[3], Steffen Thiel[4], Anne-Mette Hvas[1,2]*

1 Department of Clinical Biochemistry, Aarhus University Hospital, Aarhus, Denmark, 2 Department of Clinical Medicine, Aarhus University, Aarhus, Denmark, 3 Department of Plastic and Breast Surgery, Aarhus University Hospital, Aarhus, Denmark, 4 Department of Biomedicine, Aarhus University, Aarhus, Denmark

* am.hvas@dadlnet.dk

**Data Availability Statement:** All relevant data are within the paper and its Supporting Information files.

## Abstract

### Background

Cancer patients who undergo tumor removal, and reconstructive surgery by transfer of a free tissue flap, are at high risk of surgical site infection and ischemia-reperfusion injury. Complement activation through the lectin pathway (LP) may contribute to ischemia-reperfusion injury. Remote ischemic preconditioning (RIPC) is a recent experimental treatment targeting ischemia-reperfusion injury. The study aims were to investigate LP protein plasma levels in head and neck cancer patients compared with healthy individuals, to explore whether RIPC affects LP protein levels in head and neck cancer surgery, and finally to examine the association between postoperative LP protein levels and the risk of surgical site infection.

### Methods

Head and neck cancer patients ($n = 60$) undergoing tumor resection and reconstructive surgery were randomized 1:1 to RIPC or sham intervention administered intraoperatively. Blood samples were obtained preoperatively, 6 hours after RIPC/sham, and on the first postoperative day. LP protein plasma levels were measured utilizing time-resolved immuno-fluorometric assays.

### Results

H-ficolin and M-ficolin levels were significantly increased in cancer patients compared with healthy individuals (both $P \leq 0.02$). Conversely, mannan-binding lectin (MBL)-associated serine protease (MASP)-1, MASP-3, collectin liver-1 (CL-L1), and MBL-associated protein of 44 kilodalton (MAp44) levels were decreased in cancer patients compared with healthy individuals (all $P \leq 0.04$). A significant reduction in all LP protein levels was observed after surgery (all $P < 0.001$); however, RIPC did not affect LP protein levels. No difference was demonstrated in postoperative LP protein levels between patients who developed surgical site infection and patients who did not (all $P > 0.13$).

**Funding:** KF received funding from the Danish Cancer Society (https://www.cancer.dk/), "Helge Peetz og Verner Peetz og hustru Vilma Peetz Fond", and "Raimond og Dagmar Ringgård-Bohns Fond". AMH received funding from "Læge Sofus Carl Emil Friis og Hustru Olga Doris Friis' Legat". The funders had no role in study design, data collection and analysis, decision to publish, or preparation of the manuscript.

**Competing interests:** AMH has no conflicts of interest regarding the present paper, but have the following general conflicts for interest: has received speaker's fees from CSL Behring, Bayer, Boehringer-Ingelheim, Bristol-Myers Squibb, and Astellas and unrestricted research support from Octapharma and CSL Behring.

## Conclusions

The LP was altered in head and neck cancer patients. LP protein levels were reduced after surgery, but intraoperative RIPC did not influence the LP. Postoperative LP protein levels were not associated with surgical site infection.

## Introduction

Surgical resection of large head and neck tumors is often followed by reconstructive surgery, where the patient's own tissue, a free flap, is transferred to the surgical defect to maintain the ability to swallow, speech, and a socially acceptable appearance. Free flap reconstruction has been found to improve both functional and oncologic outcomes [1]. However, it is a major surgical intervention, and the procedure involves interruption of the blood supply to the transferred tissue resulting in ischemia of the free flap until blood supply is restored at the recipient site. Consequently, there is a risk of ischemia-reperfusion injury in the transferred tissue [2].

In recent years, the complement system has been identified as an essential contributing factor in ischemia-reperfusion injury [3–7]. The lectin pathway (LP) is the most recently described complement pathway. It can be activated by mannose-binding lectin-associated serine proteases (MASP)-1 and -2 in complex with one of the pattern recognition molecules mannose-binding lectin (MBL), H-ficolin, L-ficolin, M-ficolin or collectin liver-LK (a heteromer of CL-L1 and CL-K1) [8]. Two MBL-associated proteins (MAps), MAp19 and MAp44, with proposed complement-regulatory function, can also be found in complex with the pattern recognition molecules. Several experimental studies recognize the importance of LP activation in ischemia-reperfusion injury [6, 7, 9–11]. Furthermore, increased levels of LP proteins are reported in cancer patients [12–15] and increased pre- and postoperative MASP-2 levels have been associated with poor prognosis after cancer surgery [13, 16].

Remote ischemic preconditioning (RIPC) is an experimental treatment against ischemia-reperfusion injury, where brief periods of extremity ischemia and reperfusion are induced with an inflatable tourniquet prior to expected prolonged target tissue ischemia. Ischemic preconditioning has shown promising results as a method to attenuate ischemia-reperfusion injury in clinical and experimental studies [17–19]. The protective mechanism underlying RIPC is still largely unknown, but in healthy subjects with no ischemic event it was shown that remote ischemic preconditioning modifies the concentration of complement proteins [20, 21].

Surgical site infection occurs in 11–41% of patients undergoing free flap reconstruction of the head and neck region [22–28]. This often leads to prolonged wound healing, re-operations, and delayed adjuvant radiotherapy [23, 25]. Notably, lower LP protein serum levels have been found in cancer patients who developed postoperative infections compared with patients without infection suggesting an active role of the LP in the protection against surgical infections [29–31].

The aims of the study were to investigate: 1) LP protein plasma levels in head and neck cancer patients compared with healthy individuals, 2) whether RIPC influences LP protein levels in cancer patients undergoing free flap reconstruction, and 3) the association between LP protein levels and risk of surgical site infections. We hypothesized that: 1) LP protein levels in head and neck cancer patients are increased compared with healthy individuals, 2) RIPC induces higher postoperative LP protein levels due to decreased LP activation and LP protein tissue deposition, and 3) lower postoperative LP protein levels are associated with surgical site infection.

## Materials and methods

### Design and study population

The present study is a single-blinded, single-center, randomized controlled trial investigating the effects of RIPC in head and neck cancer patients undergoing free flap reconstruction. The effects of RIPC on the hemostatic system in the same study population have previously been published [32]. Patients were enrolled at the Department of Plastic and Breast Surgery, Aarhus University Hospital, Denmark. As previously described [32], patients were eligible for inclusion if they 1) were above 18 years of age, 2) had histologically verified or clinically suspected malignant tumor in the oral cavity, mandible, maxilla, pharynx, larynx, and/or esophagus, 3) were scheduled for tumor resection in the head and neck region followed by reconstruction with a single free flap, and 4) were able to give informed consent. Patients with arterial or venous thromboembolism within the last three months before surgery were excluded from the study. Patients were randomized 1:1 to RIPC or sham intervention during surgery. Allocation concealment and randomization was carried out as described previously [32]. Table 1 shows a summary of the clinical and perioperative characteristics of the study population.

Written informed consent was obtained from all participants prior to inclusion and the trial was carried out in accordance with the Helsinki Declaration. The study was approved by the Central Denmark Region's Ethics Committee on August 13, 2015 (journal no. 1-10-72-140-15), by the Danish Data Protection Agency (journal no. 1-16-02-358-15), and was registered with ClinicalTrials.gov (NCT02548377). The authors confirm that all ongoing and related trials for this intervention are registered. CONSORT Checklist (S1 Checklist), trial protocol (S1 File), and dataset (S2 File) are provided as supporting information.

### Intervention

Patients underwent tumor resection and free flap reconstruction under general anesthesia. For induction of anesthesia remifentanil and propofol intravenously were used. For maintenance of anesthesia sevoflurane gas was used unless contraindicated in which case propofol was used. On the day of surgery, all patients received three doses of 500 mg metronidazole intravenously and three doses of 1,500 mg cefuroxime intravenously. The following three days, all patients received 500 mg metronidazole intravenously twice daily and 1,500 mg cefuroxime intravenously three times daily. Thromboprophylaxis with low-molecular-weight heparin subcutaneously was administered pre- and postoperatively and was continued for 28 days postoperatively unless contraindicated.

RIPC was carried out intraoperatively, 35 min before expected free flap ischemia. An automated, inflatable tourniquet (autoRIC device, CellAegis Devices Inc., Toronto, Canada), was applied to the patient's upper extremity and was inflated to 200 mmHg for 5 min. This was followed by deflation of the cuff and reperfusion for 5 min. The RIPC protocol consisted of 4 cycles of 5 min ischemia and 5 min reperfusion. In the sham group the autoRIC was applied to the patient's upper extremity, but not inflated.

### Blood sampling

Blood samples were collected at three time points: prior to surgery, 6 h after RIPC/sham intervention, and on the 1st postoperative day at 06:30 AM. Blood was collected in ethylenediamine tetraacetic acid (EDTA) tubes (BD Vacutainer®, Becton, Dickinson and Company, Franklin Lakes, NJ, USA) and lithium-heparin tubes (BD Vacutainer®). All blood samples were centrifuged at 2,960 relative centrifugal force for 25 min at room temperature. Plasma was then aliquoted and stored at -80°C until analysis.

**Table 1. Demographic, clinical, and perioperative characteristics in head and neck cancer patients undergoing free flap reconstruction randomized to remote ischemic preconditioning (RIPC) or sham intervention.**

| Variable | RIPC (*n* = 30) | Sham (*n* = 30) | *P* |
|---|---|---|---|
| **Sex**, male/female, *n* (%) | 18/12 (60/40) | 19/11 (63/37) | >0.99 |
| **Age** (years) | 67 ± 10 | 64 ± 12 | 0.22 |
| **Body mass index** (kg/m$^2$) | 25 ± 4 | 23 ± 4 | 0.18 |
| **Smoking status**, *n* (%) | | | |
| Smoker | 12 (40) | 16 (53) | 0.44 |
| Non-smoker | 18 (60) | 14 (47) | |
| **Alcohol consumption**, *n* (%) | | | |
| < 21 units[a] per week | 21 (70) | 23 (77) | 0.77 |
| ≥ 21 units per week | 9 (30) | 7 (23) | |
| **ASA classification**, *n* (%) | | | |
| 1 | 1 (3) | 0 | 0.20 |
| 2 | 13 (43) | 19 (63) | |
| 3 | 16 (53) | 11 (37) | |
| **Co-morbidity** | | | |
| Charlson's co-morbidity score, median (IQR) | 5 (4–6) | 5 (4–6) | 0.98 |
| MBL concentration, preoperative, *n* (%) | | | |
| < 100 ng/ml | 3 (10) | 5 (17) | 0.71 |
| ≥ 100 ng/ml | 27 (90) | 25 (83) | |
| **Cancer status**, *n* (%) | | | |
| Tumor histology | | | |
| Squamous cell carcinoma | 25 (83) | 24 (80) | >0.99 |
| Carcinoma, other | 3 (10) | 3 (10) | |
| Osteosarcoma | 1 (3) | 1 (3) | |
| Ameloblastoma | 0 | 1 (3) | |
| No residual tumor | 1 (3) | 1 (3) | |
| Secondary malignancy | 4 (13) | 3 (10) | 0.70 |
| Neoadjuvant chemotherapy | 0 | 2 (7) | 0.49 |
| **Operative characteristics** | | | |
| Surgery time (min) | 398 ± 78 | 417 ± 95 | 0.41 |
| General anesthesia time (min) | 515 ± 66 | 518 ± 106 | 0.90 |
| Flap ischemia time (min), median (IQR) | 58 (48–95) | 58 (41–72) | 0.43 |
| Fluid balance (ml) | | | |
| Immediately postoperatively[b] | 1,953 ± 833 | 1,930 ± 720 | 0.91 |
| 1st postoperative day[c] | 2,651 ± 1,056 | 2,523 ± 1,050 | 0.64 |
| **Biochemical parameters** | | | |
| Hematocrit, preoperative (%) | 37.4 ± 4.9 | 36.6 ± 4.3 | 0.52 |
| Hematocrit, 1st postoperative day (%) | 30.5 ± 4.6 | 30.2 ± 4.1 | 0.84 |
| CRP, preoperative (mg/l), median (IQR) | 6.4 (2.7–11.8) | 3.9 (1.8–7.5) | 0.10 |
| CRP, 1st postoperative day (mg/l) | 79 ± 38 | 79 ± 30 | 0.95 |
| Leucocyte count, preoperative (x10$^9$/l) | 7.4 ± 2.4 | 7.8 ± 2.5 | 0.60 |
| Leucocyte count, 1st postoperative day (x10$^9$/l) | 11.9 ± 3.9 | 12.3 ± 3.6 | 0.64 |
| **Complications** | | | |
| Reoperation due to surgical site infection, *n* (%) | 4 (13) | 4 (13) | >0.99 |
| Antibiotics beyond three days[d], *n* (%) | 14 (47) | 14 (47) | >0.99 |

(*Continued*)

**Table 1.** (Continued)

| Variable | RIPC (*n* = 30) | Sham (*n* = 30) | *P* |
|---|---|---|---|
| 30-day mortality, *n* (%) | 1 (3) | 0 | >0.99 |

Legend: All continuous variables are presented as mean ± SD unless otherwise specified. Categorical data were analyzed with Fisher's exact test and continuous data were analyzed with unpaired t-test or Mann-Whitney test.

[a] 1 unit of alcohol = 12 g;

[b] covers the time period from start of surgery to immediately postoperatively;

[c] covers the time period from start of surgery to 06:30 AM on the 1st postoperative day;

[d] antibiotics administered for infection between postoperative day 4 and 30. ASA, American Society of Anesthesiologists; CRP, C-reactive protein; IQR, interquartile range; MBL, mannose-binding lectin; SD, standard deviation.

## Laboratory analyses

The concentrations of the LP proteins H-ficolin, M-ficolin, mannan-binding lectin (MBL), MBL-associated serine protease (MASP)-1, MASP-2, MASP-3, collectin liver-1, MBL-associated protein of 44 kilodalton (MAp44) and MAp19 were measured in EDTA plasma with time-resolved immunofluorometric assay (TRIFMA) at the Department of Biomedicine, Aarhus University, Denmark. Laboratory analyses were performed as described previously [33–38] with the exception of MASP-1 [39] and H-ficolin [40]. In brief, microtiter wells (FluoroNunc, Thermo Scientific™, Hvidovre, Denmark) were coated with specific antibodies or mannan for MBL and acetylated bovine serum albumin for H-ficolin assays. Standards, samples, and controls were added in duplicate using a Janus® Varispan automated workstation (PerkinElmer, Waltham, MA, USA). The plates were incubated with in-house biotinylated antibodies, followed by europium-labelled streptavidin (PerkinElmer), including triple washing after each step. Finally, an enhancement solution (Ampliqon, Odense, Denmark) was added and subsequently the plates were read by a fluorometer (Victor X5®, PerkinElmer).

H-ficolin was analyzed using acetylated bovine serum albumin as the capturing agent as described in [40]. MASP-1 analysis was conducted as previously described [39] except that plates were coated with mouse anti-MASP-1 antibodies (clone 1H1G6, lot no A218033050, Genscript, Piscataway, NJ, USA) at 4 µg/ml final concentration in phosphate-buffered saline as described in [40].

Each microtiter plate contained three quality controls. The inter-assay coefficient of variation (CV) was < 15% for all assays except for controls in the very low range of the standard curve for M-ficolin (CV = 32%), MASP-3 (CV = 21%), and MAp19 (CV = 18%).

For MBL, six samples from cancer patients had a plasma concentration below 10 ng/ml or a non-detectable plasma concentration. In these cases, a concentration of 10 ng/ml was used in the data analyses. For MAp19, nine samples had a low or non-detectable plasma concentration. A concentration of 225 ng/ml was used in these cases for statistical analysis, as this was the lower detection limit at the dilution used in the assay.

Plasma from healthy blood donors was collected in a previous study [39]. This group of healthy individuals served as controls for the present study (for all proteins *n* = 211 except for MASP-1, *n* = 108 and for H-ficolin, *n* = 144).

C-reactive protein (CRP) was measured in lithium-heparin plasma on Cobas® R 6000 (Roche Diagnostics, Basel, Switzerland) and leucocyte count and hematocrit value were measured in EDTA-anticoagulated whole blood on Sysmex XE-5000 (Sysmex Corporation, Kobe, Japan).

## Clinical data

Clinical data were obtained from medical records and follow-up was carried out 30 days post-operatively. Fluid balances immediately after surgery and on the 1st postoperative day at 06:30 AM were calculated. All administered fluids were registered and diuresis, perspiration, bleeding, drain output, and other fluid losses were subtracted from the administered volume to calculate fluid balance. Surgical site infection was defined as suspected abscess formation requiring re-operation under general anesthesia. Antibiotics administration during the 30 postoperative days was registered.

## Outcomes

The primary outcome was difference in MASP-2 protein concentration between the RIPC and sham group on the first postoperative day.

The secondary outcomes were: 1) difference in LP protein plasma concentrations between healthy individuals and head and neck cancer patients before surgery; 2) difference in LP protein concentration changes between the RIPC and sham group from before surgery to after surgery; and 3) difference in postoperative LP protein concentration between patients who developed surgical site infection and patients who did not.

## Sample size and statistical analysis

Plasma levels of lectin pathway proteins in head and neck cancer patients have not been described previously. However, data from a randomized controlled trial including lung cancer patients showed a mean postoperative MASP-2 plasma level of 440 ng/ml with a standard deviation of 133 ng/ml [41]. We estimated the minimal relevant difference to be 120 ng/ml. With an $\alpha$ (two-sided) of 0.05 and power $(1 - \beta)$ of 90%, we would be able to detect a statistically significant difference in MASP-2 postoperatively, if 26 patients were included in each group. We included 30 patients in each group to account for potential drop-outs or missing data.

Data were assessed for normal distribution with Q-Q plots. Data were presented with mean ± standard deviation (SD) if normally distributed, and median with interquartile range (IQR) if not normally distributed. Graphically all LP protein concentrations were presented as median with IQR as not all LP proteins were normally distributed.

Differences between groups at a single time point were analyzed with unpaired t-test for normally distributed data or Mann-Whitney test if data were not normally distributed. Welch's correction was applied for the unpaired t-test if variances differed significantly. Paired t-test was used for analysis of difference between different time points within groups. Differences in LP protein concentrations between groups over time were tested with two-way repeated measures analysis of variance (ANOVA). Categorical variables were analyzed with Fisher's exact test. Pearson's r or Spearman's r was calculated to investigate correlations between the change in hematocrit values and LP protein concentrations from the preoperative sampling time to the 1st postoperative day.

The intention-to-treat principle was used for all analyses except in the comparison of cancer patients vs. healthy individuals. Two-tailed tests were used and *P*-values < 0.05 were considered statistically significant. Prism 6.0 and 7.0 (GraphPad, La Jolla, CA, USA) was used for all statistical analyses and graphics.

## Results

Sixty patients were enrolled between September 29, 2015 and November 28, 2017; 30 patients in the RIPC group and 30 in the sham group, as described in [32]. 30-day follow-up was completed on December 28, 2017. The in- and exclusion process is shown in Fig 1.

One patient in the RIPC group died within the 30-day follow-up period after having been diagnosed with pulmonary embolism. No adverse events resulting from study interventions were reported. The two groups did not differ in demographic, clinical, or perioperative characteristics (Table 1).

### Cancer patients compared with healthy individuals

One patient had a benign tumor and two patients did not have any residual malignant disease after preoperative radiotherapy or previous surgery. These three patients were therefore excluded from the comparison of cancer patients and healthy individuals. M-ficolin and H-

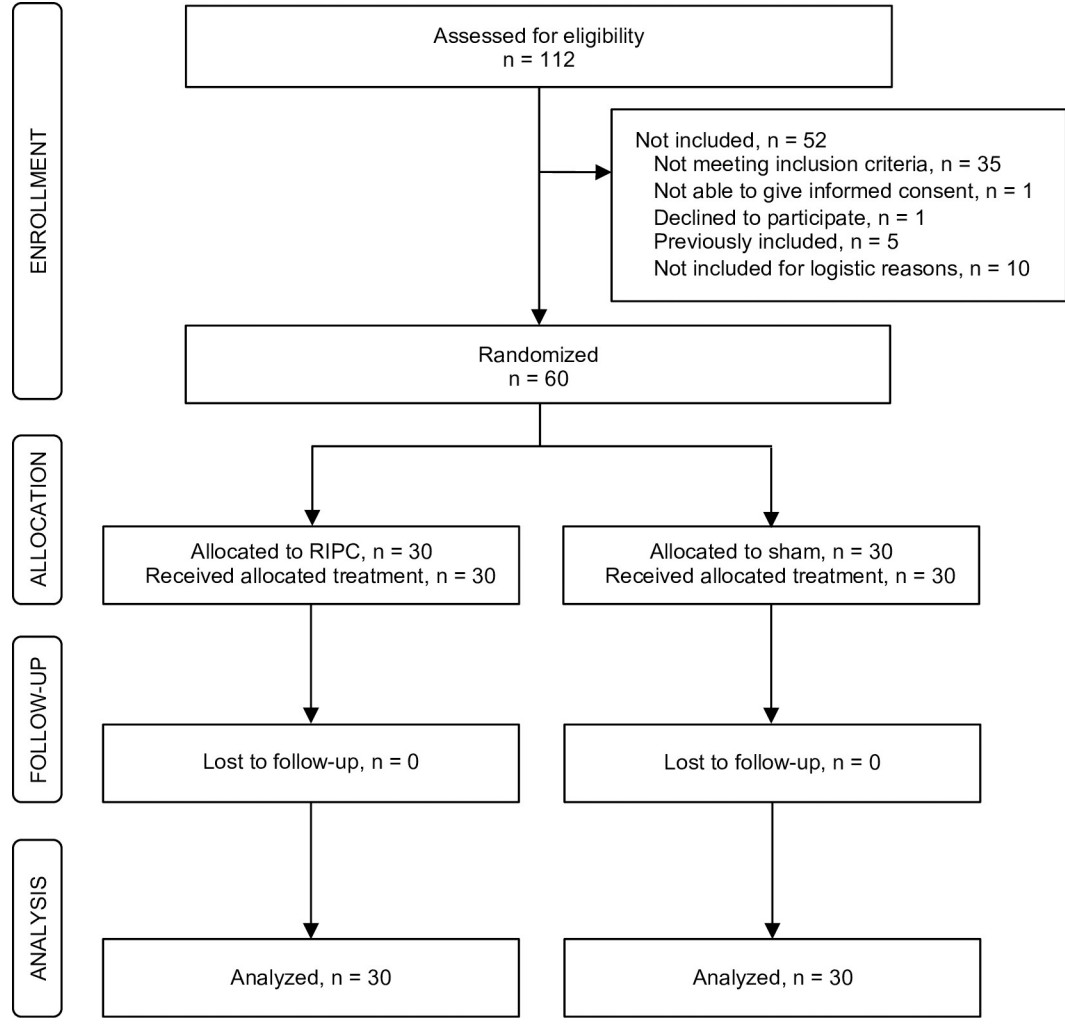

**Fig 1. Flow diagram of the study inclusion and exclusion process.** Sixty patients were randomized 1:1 to remote ischemic preconditioning (RIPC) or sham intervention. All patients received the allocated intervention, no patients were lost to follow-up and all patients were included in the intention-to-treat analyses.

ficolin concentrations were significantly higher in cancer patients compared with healthy individuals (both P-values ≤ 0.02) (Table 2 and Fig 2). Contrary to this, MASP-1, MASP-3, CL-L1, and MAp44 concentrations were significantly lower in cancer patients than in healthy individuals (all P-values ≤ 0.04). For MBL, MASP-2, and MAp19 there was no significant difference between cancer patients and healthy individuals (all P-values ≥ 0.18).

## Effect of remote ischemic preconditioning

There was no difference in MASP-2 concentrations on the first postoperative day between the RIPC and sham group (RIPC: mean 461 (SD: 212) ng/ml vs. sham: mean 490 (SD: 140) ng/ml, (P = 0.53)). There was no difference in LP protein concentrations before surgery between the RIPC and sham group except for MAp44 (RIPC: mean 1,901 (SD: 458) ng/ml vs. sham: mean 2,171 (SD: 554) ng/ml, (P = 0.04)). LP protein concentrations did not differ between the intervention and the sham group over time (ANOVA, all P-values > 0.07) (Fig 3). Further, no difference was found between the RIPC and sham group regarding the increase in CRP and leucocyte count from before surgery to the 1st postoperative day (both P > 0.54).

## Effect of surgery

As no effect of RIPC was observed, data from the RIPC and sham group were pooled (n = 60) for analysis of the effect of surgery. In this group comprising the entire study population a significant reduction in all LP protein concentrations was seen after surgery (ANOVA, all P-values < 0.001). The mean (SD) reduction from the preoperative sampling time to the 1st postoperative day ranged from 6% for CL-L1 (552 (102) ng/ml vs. 520 (105) ng/ml) to 36% for (M-ficolin 4,921 (1,840) ng/ml vs. 3,148 (1,266) ng/ml). From the preoperative sampling time to the 1st postoperative day the mean hematocrit level decreased by 18% (mean hematocrit 0.37 (SD: 0.05) vs. mean hematocrit 0.30 (SD: 0.04)). There was no correlation between the decrease in LP protein concentrations and hematocrit levels from the preoperative sampling time to the 1st postoperative day (r: (-0.24)-(0.20); all P-values > 0.07). Mean (SD) CRP increased from 11 (18) mg/l before surgery to 79 (34) mg/l on the 1st postoperative day (P < 0.0001), and mean leucocyte count (x $10^9$/l) increased from 7.6 (2.4) to 12.1 (3.7), (P < 0.0001).

**Table 2. Lectin pathway protein concentrations in healthy individuals and head and neck cancer patients prior to surgery.**

| Protein, ng/ml | (n = 211)[a] | Cancer (n = 57) | P |
|---|---|---|---|
| MBL, median (IQR) | 1,839 (574–3,009) | 1,530 (549–2,832) | 0.60 |
| H-ficolin | 20,673 ± 3,968 | 28,603 ± 7,015 | <0.0001 |
| M-ficolin | 4,304 ± 1,120 | 4,917 ± 1,888 | 0.02 |
| CL-L1 | 639 ± 160 | 549 ± 102 | <0.0001 |
| MASP-1 | 13,125 ± 3,577 | 11,801 ± 4,285 | 0.04 |
| MASP-3 | 7,202 ± 2,077 | 5,771 ± 1,847 | <0.0001 |
| MAp44 | 2,411 ± 539 | 2,024 ± 533 | <0.0001 |
| MASP-2 | 525 ± 222 | 553 ± 234 | 0.41 |
| MAp19 | 489 ± 109 | 467 ± 105 | 0.18 |

Legend: All protein concentrations are presented as mean ± SD unless otherwise specified. P-values are from unpaired t-test or Mann-Whitney test.

[a] n = 211 for healthy individuals except MASP-1 (n = 108) and H-ficolin (n = 144). CL-L1, collectin liver-1; IQR, interquartile range; MAp19, MBL-associated protein of 19 kilodalton (kDa); MAp44, MBL-associated protein of 44 kDa; MASP, MBL-associated serine protease; MBL, mannose-binding lectin; SD, standard deviation.

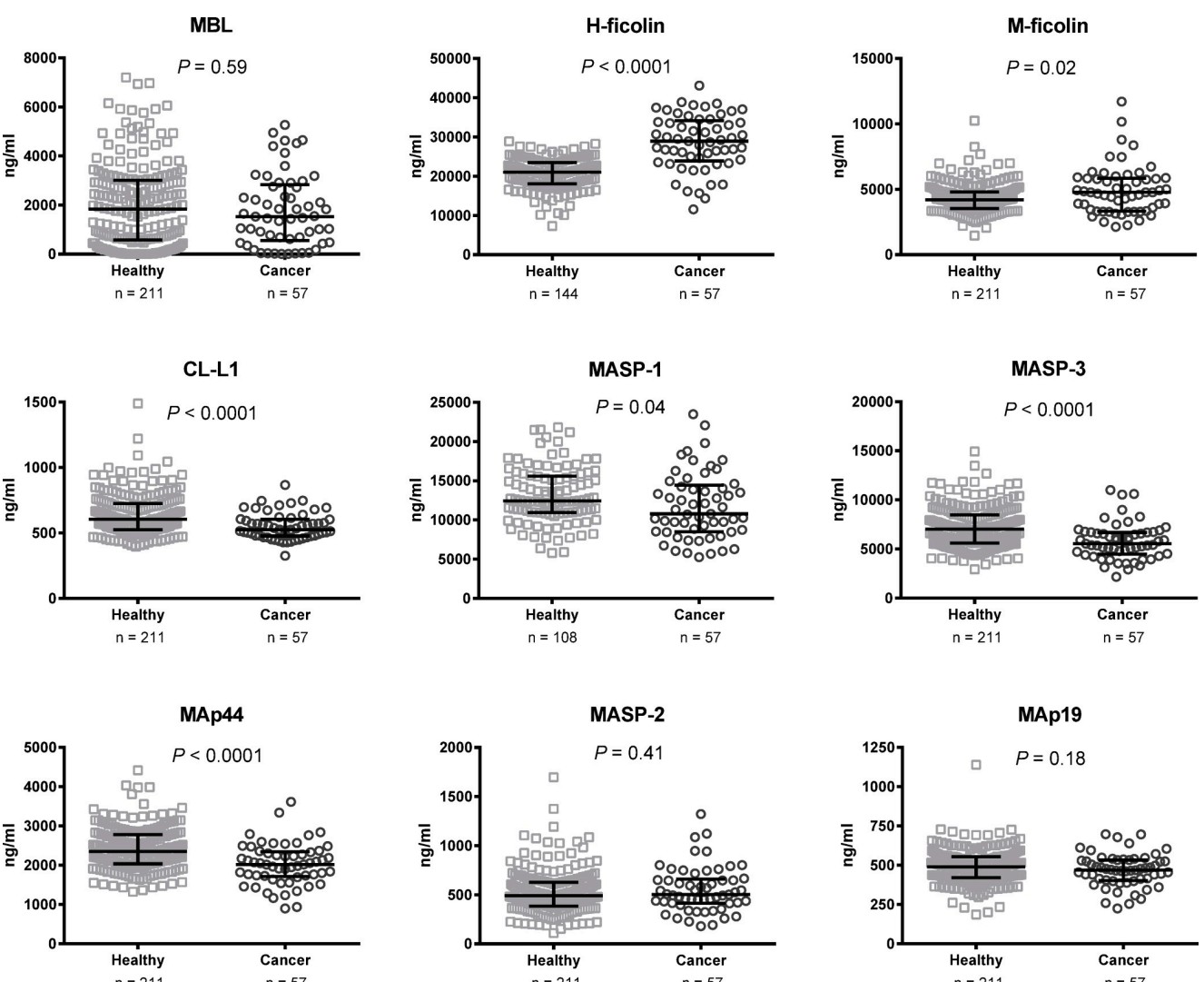

**Fig 2. Lectin pathway proteins in healthy individuals (*n* = 108–211) and cancer patients (*n* = 57) prior to surgery.** Bars indicate median and interquartile range. P-values are from unpaired t-test or Mann-Whitney test. CL-L1, collectin liver-1; MAp19, MBL-associated protein of 19 kilodalton (kDa); MAp44, MBL-associated protein of 44 kDa; MASP, MBL-associated serine protease; MBL, mannose-binding lectin.

## Surgical site infection and lectin pathway proteins

Eight (13%) patients, four from the RIPC group and four from the sham group, underwent re-operation for surgical site infection. There was no significant difference in LP protein concentrations on the 1st postoperative day between patients who were re-operated due to surgical site infection with patients who were not (all *P*-values > 0.13) (Table 3). Likewise, when comparing LP protein concentrations from the 1st postoperative day between patients who received antibiotics for infection beyond the third postoperative day (*n* = 28) with patients who did not (*n* = 32), there was no significant difference for any of the proteins (all *P*-values ≥ 0.05). As shown in Table 4, patients who consumed ≥ 21 units (1 unit = 12 g) of alcohol per week had a significantly increased risk of re-operation due to surgical site infection compared with patients who consumed < 21 units per week, (relative risk 4.6 [95% confidence interval (CI): 1.3;15.7], *P*-value = 0.03). Age, sex, body mass index, smoking status, and preoperative MBL

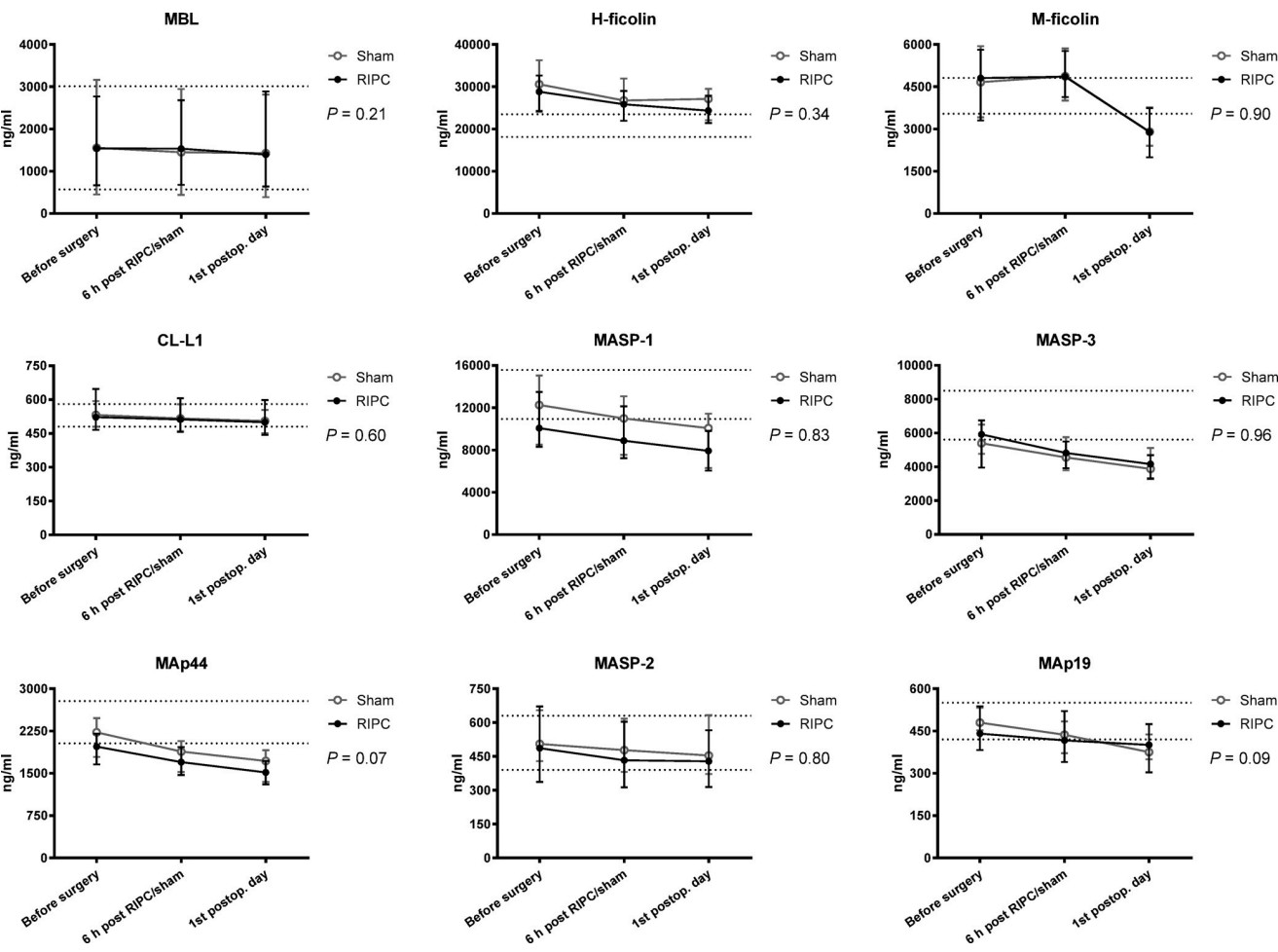

**Fig 3. Lectin pathway proteins over time in patients randomized to remote ischemic preconditioning (RIPC) (*n* = 30) or sham intervention (*n* = 30).** Bars indicate median and interquartile range. *P*-values are from two-way repeated measurements ANOVA. Dotted lines: interquartile range for the healthy individuals. CL-L1, collectin liver-1; MAp19, MBL-associated protein of 19 kilodalton (kDa); MAp44, MBL-associated protein of 44 kDa; MASP, MBL-associated serine protease; MBL, mannose-binding lectin.

concentration was not significantly different in the two groups comparing patients who were re-operated due to surgical site infection with patients who were not (all *P*-values > 0.05).

## Discussion

In summary, cancer patients demonstrated altered LP protein concentrations compared with healthy individuals. M-ficolin and H-ficolin were found in higher concentrations in cancer patients than in healthy individuals, while CL-L1, MASP-1, MASP-3, and MAp44 were found in lower concentrations in cancer patients than in healthy individuals. RIPC had no effect on LP protein concentrations in head and neck cancer patients undergoing free flap reconstruction. The surgical intervention caused a significant reduction in all LP protein concentrations. No association was found between LP protein concentrations and surgical site infection.

The finding of increased levels of M- and H-ficolin in head and neck cancer patients is consistent with previous studies of ovarian and colorectal cancer patients [14, 15, 42]. A previous study found threefold increases in H-ficolin levels in ovarian cancer patients suggesting this protein could be a potential biomarker for cancer [42]. However, a recent study by Rasmussen

**Table 3. Lectin pathway protein concentrations on the 1st postoperative day in head and neck cancer patients who developed postoperative surgical site infection (SSI) compared with patients who did not develop SSI.**

| Protein, ng/ml | SSI (*n* = 8) | No SSI (*n* = 52) | P |
|---|---|---|---|
| MBL, median (IQR) | 2,558 (687–4,489) | 1,395 (419–2,692) | 0.28 |
| H-ficolin | 27,570 ± 6,138 | 24,235 ± 5,594 | 0.13 |
| M-ficolin | 2,805 ± 1,060 | 3,201 ± 1,295 | 0.41 |
| CL-L1 | 506 ± 134 | 522 ± 102 | 0.69 |
| MASP-1 | 8,624 ± 1,970 | 8,691 ± 3,570 | 0.96 |
| MASP-3 | 4,057 ± 634 | 4,243 ± 1,293 | 0.69 |
| MAp44 | 1,628 ± 381 | 1,576 ± 362 | 0.71 |
| MASP-2 | 613 ± 266 | 454 ± 154 | 0.14 |
| MAp19 | 423 ± 93 | 397 ± 106 | 0.52 |

Legend: All protein concentrations are presented as mean ± SD unless otherwise specified. *P*-values are from unpaired t-test or Mann-Whitney test. CL-L1, collectin liver-1; IQR, interquartile range; MAp19, MBL-associated protein of 19 kilodalton (kDa); MAp44, MBL-associated protein of 44 kDa; MASP, MBL-associated serine protease; MBL, mannose-binding lectin; SD, standard deviation.

*et al.* did not find increased H-ficolin levels in patients with various types of cancer [43]. The decrease in MAp44 in the present study is consistent with earlier findings in colorectal cancer patients [14]. As previously proposed, increased levels of M-ficolin together with decreased MAp44 in cancer patients could indicate complement activation with subsequent inflammation [14]. MAp44 is thought to be a competitive inhibitor of the LP [44]. Consequently, lower levels of MAp44 might contribute to increased inflammation. M-ficolin is produced by

**Table 4. Demographic, clinical, and perioperative characteristics in patients who developed surgical site infection (SSI) compared with patients who did not develop SSI.**

| Variable | SSI | No SSI | RR [95% CI] | P |
|---|---|---|---|---|
| | (*n* = 8) | (*n* = 52) | | |
| **Sex**, *n* (%) | | | | |
| Male | 7 (88) | 30 (58) | 4.4 [0.8–26.5] | 0.14 |
| Female | 1 (13) | 22 (42) | | |
| **Age** (years) | 58 ± 11 | 66 ± 11 | | 0.05 |
| **Body mass index** (kg/m$^2$) | 23 ± 3 | 24 ± 4 | | 0.45 |
| **Smoking status**, *n* (%) | | | | |
| Smoker | 5 (63) | 23 (44) | 1.9 [0.5–6.8] | 0.45 |
| Non-smoker | 3 (38) | 29 (56) | | |
| **Alcohol consumption**, *n* (%) | | | | |
| ≥ 21 units[a] per week | 5 (63) | 11 (21) | 4.6 [1.3–15.7] | 0.03 |
| < 21 units per week | 3 (38) | 41 (79) | | |
| **MBL concentration, preoperative**, *n* (%) | | | | |
| < 100 ng/ml | 1 (13) | 7 (13) | 0.9 [0.2–4.4] | >0.99 |
| ≥ 100 ng/ml | 7 (88) | 45 (87) | | |
| **CRP, 1st postoperative day** (mg/l) | 88 ± 36 | 77 ± 34 | | 0.44 |
| **Leucocytes, 1st postoperative day** (x10$^9$/l) | 13 ± 4 | 12 ± 4 | | 0.64 |

Legend: All continuous variables are presented as mean ± SD. Categorical data were analyzed with Fisher's exact test and continuous data with unpaired t-test.
[a] 1 unit of alcohol = 12 g. CI, confidence interval; CRP, C-reactive protein; MBL, mannose-binding lectin; RR, relative risk; SD, standard deviation.

neutrophils and monocytes [45] and increased levels might therefore suggest ongoing inflammation. MBL and MASP-2 concentrations did not differ between cancer patients and healthy individuals, which is in accordance with previous studies by Rasmussen *et al.* and Swierzko *et al.* [43, 46, 47]. However, other previous studies reported increased concentrations of MBL and MASP-2 and increased MBL/MASP activity in cancer patients [12, 13, 31, 48].

Our hypothesis that RIPC induces a higher LP protein level was not confirmed. The effect of ischemic conditioning on the complement system has only been sparsely investigated previously. In healthy volunteers, RIPC predominantly caused a decrease in plasma concentrations of complement proteins [20, 21]. Conversely, a randomized controlled trial of children undergoing cardiac surgery reported increased postoperative levels of C3 and C4b in the RIPC group compared with controls [49].

In the present study, LP protein concentrations decreased significantly after surgery. Reduced perioperative levels of MBL [50], M-ficolin [33], MASP-1 [51], MASP-3 [34], MAp44 [34], and MASP-2 [50] have previously been found in cancer patients undergoing laparoscopic colectomy. However, three of these studies [33, 34, 51] only included six patients for these analyses. Moreover, an early postoperative reduction in MBL, CL-L1, H-ficolin, M-ficolin, MASP-3, MAp44, MASP-2, and MAp19, but not MASP-1, was recently reported in lung cancer patients undergoing thoracoscopic lobectomy [41]. Contrary to this, Van Till *et al.* who measured MBL concentrations several days postoperatively found increased levels of MBL on the 3$^{rd}$ and 4$^{th}$ postoperative day in patients undergoing laparoscopic esophagectomy [52]. In another study of patients with gastrointestinal cancers no effect of surgery on MBL levels was found [29]. A possible explanation behind the observed decrease in LP protein levels in the present study could be LP activation leading to increased protein deposition in damaged tissues. Both the surgical acute phase response and ischemia-reperfusion injury might be involved in this. This is supported by the observed increase in CRP and leucocyte count. In the present study, surgery induced an increase in fluid balance and a reduction in mean hematocrit levels, suggesting hemodilution. However, changes in hematocrit levels were not correlated to changes in LP protein levels. Hence, the observed reduction in LP protein levels did not seem to be explained by hemodilution. In the present study, M-ficolin showed the most pronounced perioperative change with a 36% decrease with a simultaneous increase in leucocyte count. A possible explanation for M-ficolin decrease could be exhaustion of neutrophil granulocytes, leading to a reduced M-ficolin shedding from leucocytes despite increased leucocyte counts. A similar postoperative decrease in M-ficolin was reported in a previous study [41].

No association was found between postoperative LP protein concentrations and surgical site infection or postoperative administration of antibiotics. Thus, the present study did not confirm previous findings of decreased pre- and postoperative MBL concentrations and MBL/MASP activity in cancer patients with postoperative infections [29–31]. However, only 13% of patients developed postoperative surgical site infection, which is a relatively low proportion compared with previous reports [22–28]. Post-hoc analyses revealed an increased relative risk for surgical site infection in patients with high alcohol consumption. This is consistent with the recent findings by Wagner *et al.* [24]. Low preoperative MBL concentration was not found to be associated with increased incidence of postoperative infection, which is in accordance with findings in colorectal cancer patients [31].

The major strength of the present study is the randomized controlled design. All data were collected prospectively, and no patients were lost to follow-up. Furthermore, the TRIFMA method is a highly sensitive method for measuring proteins in low concentrations and the present study included analysis of a wide range of LP proteins. Exclusively patients undergoing free flap reconstructive head and neck cancer surgery were included. The present study

therefore adds valuable novel information about this specific patient population and changes in LP proteins after cancer surgery. Some limitations should be considered. The ultimate measure of ischemia-reperfusion injury in free flap reconstruction would be loss of the transferred tissue. However, as only 3 cases of free flap failure occurred in the trial, we were not able to conduct statistical analysis on these data. The RIPC protocol used in the present study, and similar protocols with three cycles of 5 min ischemia and 5 min reperfusion, has previously been demonstrated to improve clinical and biochemical outcomes in elective surgery or other interventions in various randomized controlled trials [53–60]. However, administration of additional cycles of RIPC might enhance a potential effect of the intervention. Another possible contributing factor to the lack of effect of RIPC could be the systemic impact from the major surgery overruling a potential small effect of the intervention. Likewise, we cannot rule out that minor differences between groups have been missed due to the relatively small sample size or heterogeneity in the included patient group. Lastly, it was not possible to blind surgeons and trial investigators for the intervention, but patients and care providers were blinded.

In conclusion, the present study demonstrated that the LP was altered in cancer patients compared with healthy individuals. Intervention with RIPC did not influence LP protein concentrations after free flap reconstruction, but the surgical procedure induced a reduction in LP protein concentration. Lastly, postoperative protein concentration was not associated with surgical site infection.

## Supporting information

**S1 Checklist. CON SORT checklist.**
(PDF)

**S1 File. Trial protocol.**
(PDF)

**S2 File. Dataset.**
(XLSX)

## Acknowledgments

The authors would like to thank the head and neck microsurgical team at Aarhus University Hospital, Denmark, for supporting this study. Laboratory technicians Annette Gudmann Hansen and Lisbeth Jensen are acknowledged for their skillful laboratory assistance.

## Author Contributions

**Conceptualization:** Kristine Frederiksen, Andreas Engel Krag, Julie Brogaard Larsen, Birgitte Jul Kiil, Steffen Thiel, Anne-Mette Hvas.

**Data curation:** Kristine Frederiksen, Andreas Engel Krag, Julie Brogaard Larsen.

**Formal analysis:** Kristine Frederiksen.

**Funding acquisition:** Kristine Frederiksen, Anne-Mette Hvas.

**Investigation:** Kristine Frederiksen, Andreas Engel Krag, Julie Brogaard Larsen, Birgitte Jul Kiil, Steffen Thiel, Anne-Mette Hvas.

**Methodology:** Kristine Frederiksen, Andreas Engel Krag, Julie Brogaard Larsen, Birgitte Jul Kiil, Steffen Thiel, Anne-Mette Hvas.

**Project administration:** Kristine Frederiksen, Anne-Mette Hvas.

**Supervision:** Andreas Engel Krag, Julie Brogaard Larsen, Steffen Thiel, Anne-Mette Hvas.

**Writing – original draft:** Kristine Frederiksen.

**Writing – review & editing:** Kristine Frederiksen, Andreas Engel Krag, Julie Brogaard Larsen, Birgitte Jul Kiil, Steffen Thiel, Anne-Mette Hvas.

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
