## [Decision Letter · Decision Letter 0]

20 Jan 2020

PONE-D-19-32289

Remote ischemic preconditioning does not influence lectin pathway protein levels in head and neck cancer patients undergoing surgery

PLOS ONE

Dear Dr. Hvas,

Thank you for submitting your manuscript to PLOS ONE. After careful consideration, we feel that it has merit but does not fully meet PLOS ONE’s publication criteria as it currently stands. Therefore, we invite you to submit a revised version of the manuscript that addresses the points raised during the review process. In particular please address concerns raised as to how sample size was calculated, providing details in ths mterials and methods section, and discuss the impact of patient number and heterogeneity on results.  

We would appreciate receiving your revised manuscript by Mar 05 2020 11:59PM. To enhance the reproducibility of your results, we recommend that if applicable you deposit your laboratory protocols in protocols.io, where a protocol can be assigned its own identifier (DOI) such that it can be cited independently in the future. For instructions see: http://journals.plos.org/plosone/s/submission-guidelines#loc-laboratory-protocols

We look forward to receiving your revised manuscript.

Kind regards,

Ivan R. Nabi, Ph.D.

Academic Editor

PLOS ONE

Journal Requirements:

2. Thank you for submitting your clinical trial to PLOS ONE and for providing the name of the registry and the registration number. The information in the registry entry suggests that your trial was registered after patient recruitment began. PLOS ONE strongly encourages authors to register all trials before recruiting the first participant in a study.

i) your reasons for your delay in registering this study (after enrolment of participants started);

ii) confirmation that all related trials are registered by stating: “The authors confirm that all ongoing and related trials for this drug/intervention are registered”.

Please also ensure you report the date at which the ethics committee approved the study as well as the complete date range for patient recruitment and follow-up in the Methods section of your manuscript.

3. We noticed you have some minor occurrence(s) of overlapping text with the following previous publication(s), which needs to be addressed:

https://doi.org/10.1371/journal.pone.0219496

https://doi.org/10.1111/aji.13092

https://doi.org/10.1111/sji.12680

In your revision ensure you cite all your sources (including your own works), and quote or rephrase any duplicated text outside the Methods section. Further consideration is dependent on these concerns being addressed.

Reviewers' comments:

Reviewer's Responses to Questions

**Comments to the Author**

1. Is the manuscript technically sound, and do the data support the conclusions?

Reviewer #1: Yes

Reviewer #2: Yes

2. Has the statistical analysis been performed appropriately and rigorously? 

Reviewer #1: N/A

Reviewer #2: Yes

3. Have the authors made all data underlying the findings in their manuscript fully available?

Reviewer #1: Yes

Reviewer #2: Yes

4. Is the manuscript presented in an intelligible fashion and written in standard English?

Reviewer #1: Yes

Reviewer #2: Yes

5. Review Comments to the Author

Reviewer #1: I appreciate the opportunity to review your revised manuscript. The topic is of Novelty and of clinical importance. （P refers to page, L referes to line, for examole,P10L191-196 refers to Page 10 line 191-196）.

P10L191-196, “The outcomes were: 1) ……”. In clinical trial, there are primary outcome and with or without secondly outcomes, the primary outcome is usually the most interested outcome and relative to sample calculation. Therefore, I would suggest pointing out which outcome was primary outcome.

P10L199-200, “A sample size calculation was conducted for the study on hemostasis[32], which defined the number of participants in the present study ”. It seems that the sample size was calculated the same as the article mentioned. But I read the article mentioned , and find the sentence，“The sample size was calculated with collagen-induced platelet aggregation (COL test，Roche Diagnostics) on the first postoperative day as primary endpoint”，since the outcomes were totally not the same in the two trial, I don’t think the sample size calculation could be the same. So, to make it clere that how sample size was calculated would be more appropriate.

Reviewer #2: Interesting paper dealing with "Remote ischemic preconditioning" and lectin pathway protein levels in

head and neck cancer patients undergoing surgery.

While it is a good idea to use ischemic preconditioning in these patients no effects were observed.

Might be due to heterogeneous patients or small patient number though. The authors should emphasize this a bit more in the introduction and discussion section.

6. PLOS authors have the option to publish the peer review history of their article (what does this mean?). If published, this will include your full peer review and any attached files.

Reviewer #1: No

Reviewer #2: No

---

## [Author Response · Author response to Decision Letter 0]

14 Feb 2020

Our response letter is uploaded as a file.

---

## [Editor Report · Decision Letter 1]

2 Mar 2020

Remote ischemic preconditioning does not influence lectin pathway protein levels in head and neck cancer patients undergoing surgery

PONE-D-19-32289R1

Dear Dr. Hvas,

We are pleased to inform you that your manuscript has been judged scientifically suitable for publication and will be formally accepted for publication once it complies with all outstanding technical requirements.

With kind regards,

Ivan R. Nabi, Ph.D.

Academic Editor

PLOS ONE
---

## [Editor Report · Acceptance letter]

25 Mar 2020

PONE-D-19-32289R1 

Remote ischemic preconditioning does not influence lectin pathway protein levels in head and neck cancer patients undergoing surgery 

Dear Dr. Hvas:

I am pleased to inform you that your manuscript has been deemed suitable for publication in PLOS ONE. Congratulations! Your manuscript is now with our production department. 

With kind regards,

on behalf of

Dr. Ivan R. Nabi 

Academic Editor

PLOS ONE